# Implementation and Adherence of a Custom Mobile Application for Anonymous Bidirectional Communication Among Nearly 4000 Participants: Insights from the Longitudinal RisCoin Study ^[note 2]^

**DOI:** 10.3390/idr17040088

**Published:** 2025-07-24

**Authors:** Ana Zhelyazkova, Sibylle Koletzko, Kristina Adorjan, Anna Schrimf, Stefanie Völk, Leandra Koletzko, Alexandra Fabry-Said, Andreas Osterman, Irina Badell, Marc Eden, Alexander Choukér, Marina Tuschen, Berthold Koletzko, Yuntao Hao, Luke Tu, Helga P. Török, Sven P. Wichert, Thu Giang Le Thi

**Affiliations:** 1Institut für Notfallmedizin und Medizinmanagement (INM), LMU Klinikum, LMU München, 80336 Munich, Germany; 2Department of Pediatrics, Gastroenterology and Nutrition, School of Medicine Collegium Medicum, University of Warmia and Mazury, 10-561 Olsztyn, Poland; 3Department of Pediatrics, Dr. von Hauner Children’s Hospital, LMU University Hospital Munich, 80337 Munich, Germany; 4Department of Psychiatry and Psychotherapy, University of Bern, 3000 Bern, Switzerland; 5Institute of Psychiatric Phenomics and Genomics (IPPG), LMU University Hospital Munich, 80336 Munich, Germany; 6Department of Psychiatry and Psychotherapy, LMU University Hospital Munich, 80336 Munich, Germany; 7Department of Neurology, LMU University Hospital Munich, 81377 Munich, Germany; 8Department of Medicine II, LMU University Hospital Munich, 81377 Munich, Germany; 9Max von Pettenkofer Institute and Gene Center, Virology, National Reference Center for Retroviruses, LMU Munich, 80336 Munich, Germany; 10German Center for Infection Research (DZIF), Partner Site, 80336 Munich, Germany; 11KAIROS GmbH, 44801 Bochum, Germany; 12Department of Anesthesiology, Laboratory of Translational Research Stress and Immunity, LMU University Hospital Munich, 81377 Munich, Germany; 13German Center for Child and Adolescent Health, Site Munich, 80337 Munich, Germany; 14Stiftung Kindergesundheit, c/o Dr. von Hauner Children’s Hospital, LMU University Hospital Munich, 80337 Munich, Germany

**Keywords:** cohort study, COVID-19, mobile app, health care worker, inflammatory bowel disease

## Abstract

**Background**: The longitudinal RisCoin study investigated risk factors for COVID-19 vaccination failure among healthcare workers (HCWs) and patients with inflammatory bowel disease (IBD) at a University Hospital in Germany. Since the hospital served as the study sponsor and employer of the HCW, we implemented a custom mobile application. We aimed to evaluate the implementation, adherence, benefits, and limitations of this study’s app. **Methods**: The app allowed secure data collection through questionnaires, disseminated serological results, and managed bidirectional communication. Access was double-pseudonymized and irreversibly anonymized six months after enrollment. Download frequency, login events, and questionnaire submissions between October 2021 and December 2022 were analyzed. Multivariable logistic regression identified factors associated with app adherence. **Results**: Of the 3979 participants with app access, 3622 (91%) used the app; out of these, 1016 (28%) were “adherent users” (≥12 submitted questionnaires). App adherence significantly increased with age. Among HCW, adherent users were more likely to be non-smokers (*p* < 0.001), working as administrators or nursing staff vs. physicians (*p* < 0.001), vaccinated against influenza (*p* < 0.001), and had not travelled abroad in the past year (*p* < 0.001). IBD patients exposed to SARS-CoV-2 (*p* = 0.0133) and those with adverse events following the second COVID-19 vaccination (*p* = 0.0171) were more likely adherent app users. Despite technical issues causing dropout or non-adherence, the app served as a secure solution for cohort management and longitudinal data collection. **Discussion:** App-based cohort management enabled continuous data acquisition and individualized care while providing flexibility and anonymity for the study team and participants. App usability, technical issues, and cohort characteristics need to be thoroughly considered prior to implementation to optimize usage and adherence in clinical research.

## 1. Introduction

Mobile applications (apps) are increasingly utilized in basic and applied clinical research [1]. They provide an interactive, agile platform for data collection on patient-related outcomes, disease management, and medication adherence [2,3]. The majority of previously described app implementations in clinical research report on designs mainly intended for unidirectional, not or not fully anonymized patient-reported data documentation, where the digitalization of previously paper-based methods is often the primary aim [2,3,4]. However, the use of apps for the management of a study cohort remains rarely reported [5].

The study “Risk Factors for COVID-19 Vaccine Failure” (RisCoin) was a research project by and at the Ludwig Maximilian University (LMU) Hospital in Munich, with implementation of an app specifically adapted to prospectively monitor a large cohort (Clinical trial registration: “Risk factors for COVID-19 vaccine failure.—RisCoin”, MainID: DRKS00026804, http://drks.de/search/en/trial/DRKS00026804 (accessed on 18 July 2025)) [6]. The need for a customized app solution emerged from the study design, data protection, and management concept, which included communication of laboratory results to individual participants and longitudinal collection of self-reported data. Follow-up included short questionnaires per app regarding breakthrough infections after completion of basic COVID-19 immunization, symptoms of suspected infection, or vaccination updates. Blood sampling was performed at enrollment and during follow-up to measure anti-spike (anti-S), anti-nucleocapsid (anti-N), and neutralizing antibodies against SARS-CoV-2, with individual results anonymously communicated to the participants via the study’s app [6]. The adopted approach required particular stringency as the LMU University Hospital served as both a sponsor of the study and an employer of the participating healthcare workers (HCWs) [6].

This study aims to evaluate the implementation, adherence, effectiveness, and limitations of a custom mobile app designed for anonymous bidirectional communication in a large cohort of healthcare workers and patients with IBD within the RisCoin study [6]. In this manuscript, we present the technical considerations, implementation, advantages, and drawbacks of the app used in the RisCoin Study, demonstrating its feasibility for app-based study and participant management. We discussed our experience on the acceptance and adherence to the app within the RisCoin cohort, along with the technical and managerial benefits and limitations encountered during the study.

## 2. Materials and Methods

### 2.1. Concept and Architecture of the RisCoin Study App on CentraXX App

The development and implementation of the app complied with best practices for human subject protection, as well as the quality management requirements for medical products [7]. Incorporating the app into the RisCoin study fulfilled multiple data protection requirements by operating without any personal information. The app served as an interface that enabled secure communication between participants and the RisCoin team without disclosing any personal identifiers, while still encompassing all individual study results. In addition, the app allowed participants to anonymously submit longitudinal data on their vaccination and infection status. Moreover, all patient-related data in a patient-centric database were hosted at the LMU University Hospital. A flowchart of the RisCoin cohort is available in Appendix A.

Participants could access the system and their data transparently via a slimmed-down web interface (desktop version) or the dedicated app, available for Android and iOS smartphones (Appendix A).

### 2.2. Technical, Ethical, and Data Protection Considerations of CentraXX App

The key features of the app were designed specifically for the requirements of the RisCoin study in collaboration with KAIROS GmbH. The development of a custom app instead of the adoption of ready-to-use solutions stemmed from the data protection requirements of the study’s design.

The development and implementation of the app complied with best practices for human subject protection, as described by Rosa et al. [7]. KAIROS developed the CentraXX app according to the guidelines and under a quality management system adhering to DIN EN ISO 13486, which is applied to the building of medical products [8]. Several standard operating procedures (SOPs) and working instructions were followed to ensure quality and safety in terms of building a patient-facing product. The system was based on the principles of GMP/GAMP5 (Good (Automated) Manufacturing Process) and characterized by SOPs for the analysis and design of feature requirements, as well as for different testing methods according to a specified risk management, which can also be continuous, automated, and monitored [9]. Requests and bugs were tracked and subsequently entered into the software development lifecycle again to improve the products and maintain the correct functionalities and flawless usage. The detailed protocol and guidance for the functions of the app were submitted to and approved by the Ethics Committee of the LMU Munich [6]. As further described by Koletzko et al., the CentraXX database used for the study was only accessible to a limited number of staff with limited role-based rights in their study profiles [6]. Participants’ identities were confirmed during onboarding through submitted and signed informed consent forms, and participants were trained in using the app. Additionally, the app was regularly updated following feedback and suggestions from the study team and by the participants [10]. The app communicated via standardized FHIR (Fast Health Interoperability Resources) to read and write data from and to the CentraXX server instance. Encryption via TLS (Transport Layer Security), a 2-factor authentication, and specific access rights and roles maintain a high level of security that is needed for sensitive data.

The app was developed as a native app on iOS (Swift 6, Apple Inc., Cupertino, CA, USA) and Android (Kotlin 2.0.0, JetBrains, Amsterdam, The Netherlands). A cross-platform development approach (e.g., React Native or Flutter, where a large amount of code is just written once for all platforms) was not feasible due to security and performance reasons. Platform differences were actively embraced when it came to the User Interface and User Experience because the different Human Interface guidelines from each platform manufacturer were taken into account [11,12]. The development of the app was accomplished within two months (August–September 2021).

CentraXX app was used under the institutional license of LMU University Hospital, which sponsored and hosted the RisCoin study. We used CentraXX to store all patient-related data in a patient-centric database hosted at the LMU Hospital. The server application was accessible to the study personnel and participants. CentraXX acted as a “single source of truth”, gathering all relevant data in one place to not only avoid double data entry or data duplication but also to benefit from dealing with only one application for all study purposes.

### 2.3. Administration of the RisCoin Study Participant Profile

The overall data protection concept has been previously described [6]. It was approved by the LMU data protection officer (15 September 2021), including the amendment for the second follow-up (8 September 2022). The Ethics Committee of the LMU Munich approved the study protocol (21 September 2021, Project Number: 21-0839) and its subsequent amendments (22 February 2022, 4 May 2022). Briefly, all data and biomaterials were double-pseudonymized during recruitment, and the first follow-up, and then irreversibly anonymized six months after the study started [6]. Prior to fully irreversible anonymization in July 2022, identity logs (ID logs) with RisCoin-ID and individual participant identifiers were stored externally by an independent Trusted Third Party in order to allow participant identification, e.g., in cases of participants who have lost their admission code or their password [6]. (Figure 1). Further details on the data management process are presented in the publication on the design, methods, and participants of the RisCoin study [6].

Only healthcare workers (HCWs) and adult patients with inflammatory bowel disease (IBD) (≥18 years of age) received access to the app, excluding patients with psychiatric disorders to participate in the follow-up (Appendix A).

### 2.4. User Journey and Messaging Functions

During onboarding, each participant was randomly assigned to a RisCoin study app profile, including Contact-ID, a unique number code, and a QR code for app activation matched with their Contact-ID. Participants were identifiable to the study team only via the Contact-ID.

Participants who did not, for any reason, use the mobile app could access the short questionnaires, all information, results, and services through the desktop version (Appendix A). The section “Measurements” provided all serological results of the participant, including their interpretation (Appendix A).

#### 2.4.1. Mass Messaging

Mass messages were sent in cases of updates relevant to all participants, e.g., communicating newly uploaded results, reminding participants to fill out the initial or short questionnaires, and informing them of follow-up serological controls.

#### 2.4.2. Hotline

The hotline function allowed bidirectional messaging between participants and the team. All mass and bidirectional hotline messages appeared in the same subsection of the app (“Messages”). Participants were able to ask questions, seek support, or report changes in their infection and vaccination status.

#### 2.4.3. Short Questionnaire

A short questionnaire collected data on possible or confirmed infection, vaccination updates, including adverse events, COVID-19-related symptoms, and analgesic or antipyretic intake after enrollment. We asked participants to fill out the questionnaire once a week, or whenever their infection or vaccination status had changed (event-related).

### 2.5. Data Extraction and Analysis

First, we reported the app downloads statistics from the Play Store and Apple Store (digital distribution services by Google, respectively, Apple) between 1 October 2021 and 31 December 2022 (the last day of the second follow-up). We documented the number and timing of updates.

Our second dataset, exported from the CentraXX app, comprised login event frequencies and their purposes. Data were verified for duplicate entries and plausibility. Based on this data source, we illustrated the cumulative frequency of key components in participant monitoring, including updates on vaccination status, test results, symptoms, and medication usage.

Next, we merged the dataset containing the frequency of short questionnaire submissions with data from the initial RisCoin questionnaire, including demographics, underlying diseases, medication intake, diet, lifestyle, and self-perceived stress (assessed and presented as an overall score using the validated Perceived Stress Questionnaire (PSQ) [13,14,15]). We defined active users as participants completing ≥1 short questionnaires. Participants who submitted ≥12 questionnaires (75th percentile of the cohort) were classified as adherent users. The threshold of ≥12 questionnaires was further determined based on the cumulative number of entries (Figure 1), displaying that the majority of participants used the app continuously for 3 months after enrollment. On average, participants submitted approximately one entry per week, indicating adherence to the app during the first 12 weeks after enrollment. These observations support the consideration of 12 entries as a reasonable threshold to distinguish between adherent and non-adherent users. Nevertheless, this approach has been acknowledged as a limitation of the study and discussed accordingly in the manuscript.

We performed Pearson’s Chi-square test to explore differences between adherent vs. non-adherent users, as well as active users vs. non-app users. We conducted a multivariable logistic regression including all factors associated with a high likelihood for frequent usage (*p*-value ≤ 0.25). Applying backward elimination and adjusted for gender and age (years), the final logistic regression was presented with no missing co-variates. Estimated odds ratio (OR) and 95% confidence interval (CI) with respective *p*-values obtained from the Wald Chi-square test were reported.

All analyses were conducted using SAS 9.4 (Statistical Analysis Software, SAS Institute Inc., Cary, NC, USA).

## 3. Results

### 3.1. App Usage

Overall, 8027 app downloads were recorded, with 5580 from the App Store. The ratio between iOS and Android users was approximately 1.2 to 1. During the study (7 October 2021–31 December 2022), the number of app downloads and login events was high during recruitment (7 October–16 December 2021) and remained at high levels in the first quarter of 2022 (Figure 1). For the remaining months of 2022, the app usage in terms of downloads and login events stagnated and, subsequently, gradually declined (Figure 1).

### 3.2. Hotline

Between October 2021 and December 2022, the RisCoin team recorded 1964 bidirectional communications with 958 participants. Of those, 25 participants used the hotline to report a PCR-confirmed SARS-CoV-2 infection, and 89 reported a booster vaccination (*n* = 85, 3rd vaccination, *n* = 8, 4th vaccination, *n* = 3, 5th vaccination).

### 3.3. User Statistics Through Short Questionnaire and Hotline Module

The full dataset from the questionnaire consisted of 43,369 records without duplicates. Of these, 42,546 records, produced by 3814 users, were complete (Figure 2). The user statistics presented here reflect the reporting behavior of the participants via the short questionnaire and bidirectional communications in the hotline module.

Participants reported mostly on their SARS-CoV-2 testing and results, with 3135 users having reported at least one test. Booster vaccinations (third dose) were reported by 2674 of 3814 users with two vaccinations at enrollment, and 162 participants disclosed their fourth dose. Prevalence of SARS-CoV-2 symptoms and related intake of analgesics or antipyretics were reported at least once by the majority of users (*n* = 2062; 68.2%) (Figure 2).

### 3.4. Active User Rate, Adherence, and Associated Factors

Of the 3979 RisCoin participants with app access, 3622 (91%) were active users with at least one submitted short questionnaire and 357 participants; therefore, 337 HCWs had not submitted a single entry. Among HCW, non-users were more likely to be males compared to females (11% vs. 8%, *p* = 0.0024), working full-time as physicians, and reporting high PSQ scores (Appendix A). The multivariable logistic regression analysis confirmed the following factors significantly associated with non-use of the app: being employed as a physician in a clinical setting, older age, lack of exposure risk to SARS-CoV-2, not having received an influenza vaccination during the last flu season, and higher perceived stress scores (Appendix A). Additionally, 20 IBD patients did not submit any entries via the app, but the small size of this subgroup did not allow further association studies. (Appendix A).

Table 1 presents characteristics of adherent compared to non-adherent app users among HCWs and patients with IBD. A larger proportion of IBD patients were adherent users compared to HCWs (40% vs. 27%, respectively). Both sub-cohorts presented significant differences in app adherence based on age and COVID-19 exposure. Among HCWs, a larger proportion of the adherent user group was female than male (30% vs. 20%, *p* < 0.001), older >50 years than younger (41% of 51–60 years, 48% of ≥60 years, *p* < 0.001), part-time employed (33% vs. 25%, *p* < 0.001), and with middle school education (37% vs. 23% with university degree, *p* < 0.001). Adherent users in the IBD patient sub-cohort were more present among the ≥50 years age groups (54% of 51–60 years old, 53% of ≥60 years, *p* = 0.029) and those who had been in contact with a SARS-CoV-2-positive person (58%, *p* = 0.049) or had experienced adverse events following their second COVID-19 immunization (52%, *p* = 0.047).

Among HCW, adherent app usage was significantly associated with increasing age, e.g., participants older than 60 years were 5.25 times more likely to have made ≥12 entries than participants aged 30 or younger (95% CI: 3.79–7.27, *p* < 0.0001; Figure 3A). Women (OR = 1.64, 95%CI: 1.34–2.02, *p* < 0.0001), participants who had received an influenza vaccine (OR = 1.34, 95%CI: 1.12–1.62, *p* = 0.0018), those with a pollen allergy (OR = 1.29, 95%CI; 1.09–1.54, *p* = 0.0035) or following a special diet (OR = 1.42, 95%CI: 1.06–1.96, *p* = 0.0189), and non-smokers (OR = 1.48, 95%CI: 1.19–1.86, *p* = 0.0006) were significantly associated with higher chance of being a frequent app user. Compared to physicians, all other occupational groups were more likely adherent app users, with an OR of 2.17 for participants working in administration (95% CI: 1.63–2.88, *p* < 0.0001). Participants who travelled abroad in the previous year or those experiencing above-average stress levels (PSQ score > 33) showed reduced adherence to the app [13,15]. We did not find significant differences between HCWs with and without reported diseases or regular medication intake (Figure 3A).

Among patients with IBD, increasing age, but not gender, was associated with adherent use of the app (e.g., patients aged >60 years almost 12 times more likely to be adherent users than patients <30 years of age, Figure 3B). Contact with a SARS-CoV-2 positive person (OR = 3.73, 95%CI: 1.32–10.57, *p* = 0.0133) as well as experiencing mild to moderate symptoms following the second COVID-19 vaccination (OR = 2.73, 95%CI: 1.20–6.24, *p* = 0.0171) were significantly associated with increased app adherence.

## 4. Discussion

Here, we reported our experience and quantifiable data by conducting a large-scale longitudinal prospective study that managed and followed up irreversibly anonymized HCWs and IBD patients using a mobile application. Our large cohort allowed us to identify participants’ characteristics associated with app adherence and revealed women, older participants, and patients as reliable groups adhering to app usage. The strict data protection incorporating a custom-designed mobile app could serve as a blueprint for future research requiring complete and irreversible anonymization.

Almost all participants were willing to download the app at enrollment; however, 9% of the HCWs and 12% of the patients with IBD did not use it. Adherence was particularly high among female HCWs and those compliant with annual influenza vaccination, following dietary restrictions, or non-smoking. Patients with IBD were more likely to use the app frequently if they had been exposed to a person with confirmed SARS-CoV-2 infection or experienced an adverse event to their second COVID-19 vaccination. Notably, app adherence was significantly related to increasing age in both cohorts.

Nowojewski et al. observed a similar age-related trend [4]. Still, in the broader context of telemedicine, age has been identified as a barrier to adoption [17]. In our experience, once older participants received proper guidance, they proved to be reliable in using the app independently. As the age group of 18–30 years was the largest among app users while simultaneously demonstrating the lowest adherence, the participants’ behavior may have affected the data collection due to underreporting. Concerning cohort studies in general, the participation and adherence patterns of age groups need to be actively approached during the study design phase, e.g., by planning for additional personnel to support older participants and arranging incentives for younger participants to facilitate long-term compliance.

The significantly better adherence to the app and timely data provision in females compared to male participants is in accordance with published literature reports systematic gender effect on online survey participation and non-response, including scientific studies [18]. The higher adherence of women is often explained by gender-specific views on social exchanges on digital platforms, as well as on differences in decision-making processes when considering potential benefits of participation vs. non-participation [18,19,20,21]. Still, no scientific consensus on the origins of the differences exists [18].

The higher adherence among IBD patients compared to HCWs could stem from the different purposes for which apps are typically designed for these two groups. While apps designed for HCWs mainly focus on patient and hospital administration, patient apps primarily aim to facilitate disease management [22,23]. Additionally, patients with IBD undergoing immunosuppressive therapy likely perceived a higher risk during the pandemic, concerning acquiring infection and experiencing severe disease outcomes [15]. A recent publication from the RisCoin study demonstrated impaired humoral responses to COVID-19 mRNA vaccination in patients with IBD receiving immunosuppressive therapies targeting tumor necrosis factor-alpha compared to other commonly used biologics for IBD [24]. Having a chronic condition has been previously identified as a predictor for digital health engagement among internet users in Germany, which may have contributed to the higher adherence of IBD patients compared to HCWs [25]. The fact that IBD patients were recruited for the study via the outpatient clinic through their attending physicians may also have contributed to better adherence [26].

Among HCWs, physicians were less likely to be adherent users than any other occupational group, while participants working in administration had the highest likelihood of app adherence. This discrepancy exists despite the physicians’ active involvement in and awareness of clinical research at an academic hospital. However, the workload during the pandemic and the overtime were particularly high for physicians and may be the explanation for the higher rate of non-users.

Other factors associated with app adherence among HCWs were health-related determinants, potentially related to HCWs’ health literacy. Among these, non-smoking and influenza immunization have previously demonstrated associations with increased prevention measures uptake [27,28,29,30]. Paradoxically, preceding RisCoin analyses indicated a decreased SARS-CoV-2 infection rate among frequent (daily) smokers; however, the exact mechanism of association remains unclear [31]. The reported results in regard to smoking habits reflect on previously published evidence as Nilan et al. report on 21% smoking prevalence among HCWs in high-income countries (comparable to the 18% rate observed in our cohort) and particularly high among male HCWs working as physicians, thus reflecting the non-adherent user profile identified here [32]. The same analyses found a significantly increased infection rate among participants who had travelled abroad more than four times in the previous year, while travel behavior was inversely associated with adherent app usage here [31]. Travel behavior, especially international travel, was affected significantly by restriction policies, as well as by the associated infection risk, leading to substantial changes in the travel decision-making process. As Williams et al. report on international tourism travel specifically, tolerance of situational and general risk, as well as the perceived competence to manage these and the uncertainty, e.g., regarding future restrictions, are significantly associated with people proceeding with instead of delaying a vacation abroad, thus potentially increasing their individual infection exposure [33]. With respect to the data collected within the RisCoin study, we were not able to differentiate between tourism- and non-tourism-related travel, such as participation in congresses or traveling related to family issues. The joint consideration of these associations poses a future research question on the interaction between travel behavior, infection risk, and adherence to longitudinal monitoring measures.

Additional circumstantial factors need to be considered as impediments to app adherence. The onboarding process took place in the fast-paced environment of the hospital’s vaccination center [34]. This may have led to some participants not understanding the team’s instructions. Despite providing further information and support, we cannot exclude a certain limit in adherence due to insufficient understanding. Moreover, despite highlighting the importance of safekeeping the welcome letter, several participants lost their access data, thus either enforcing re-boarding or becoming lost to follow-up [6].

The results need to be considered against the background of the study being open-ended due to the unforeseen pandemic development. We were unable to inform participants about how long they would need to fill out the questionnaire, leading to the presumption that some had stopped filling out the questionnaire either unintentionally or whenever they had seen as appropriate.

We also note that the communication of the first serological results (October 2021) was delayed and was evidently followed by a surge of dissatisfaction. This may have demotivated participants to further engage with the study.

### 4.1. Pitfalls and Frequent Issues of an App-Based Study Design

#### 4.1.1. Usability Issues

Although the app had a minimalistic and user-friendly design, we observed some difficulties in navigating it among users. A notable issue arose from the differences in operational systems and, particularly, the necessary iOS updates due to repeated crashes, which required reinstallation. This often-demanded on-site support from the study team. The higher Apple Store downloads reflect several iOS-specific updates necessitating complete reinstallation (Figure 1).

The lack of push notifications, which although necessary, impeded the timely communication with participants. Despite the team’s efforts to employ all possible platforms (hospital’s website, newsletter, posters), the inability to alert participants about new messages accounts for a certain proportion of the non-adherence to the app and the loss-to-follow-up. Indeed, future app-based data collection efforts may benefit from including a more participatory approach in facilitating user experience and, consequently, facilitating app and study protocol adherence [35,36].

#### 4.1.2. Validity and Reliability of the Data

We occasionally encountered inconsistencies in participant-reported data, e.g., conflicting information about the date and type of vaccinations. This may have been caused by typing or self-reporting errors. Such issues could have been avoided with an automatic plausibility check of previously submitted questionnaires and delivering an error message to alert users of unnecessary repetitions, e.g., delivering an error message if a participant attempts to submit a second report on their third COVID-19 vaccination. These potential solutions may be useful in other study designs. However, compared with paper-based diaries, online-based options have shown to have fewer missing data scores and errors [37]. Data dissonance proved to be particularly challenging during the data revision and cleaning process, as inconsistent data had to be harmonized in order to decrease potential missing scores. However, compared with paper-based diaries, online-based options have been shown to have fewer missing data scores and errors [27]. Additionally, a paper-based solution would not have allowed for the longitudinal and exhaustive data collection achieved with the RisCoin app. Future prospective cohort studies may be able to provide insights into the advantages and disadvantages of paper- vs. online-based data collection for the study team, as well as for participants, by employing a randomized design.

### 4.2. Technical Issues Unrelated to Study Design

#### 4.2.1. Older Phones Incapable of Supporting the App

Several technical characteristics of the study cohort impeded the rollout of the app. First, some participants did not have smartphones or their smartphones were not capable of downloading and supporting the app. All participants received the information, as well as their respective login data for the alternative desktop solution, and were enabled to use it. Some participants opted for the desktop instead of the mobile version of the app due to individual safety concerns regarding the download of apps from either app store. We were unable to report the number of participants using the desktop instead of the mobile version, as the simultaneous use of both was possible and available to all.

#### 4.2.2. Differences Between Operational Systems

A further challenge originated from the app development on two platforms, which led to noticeable differences in the functioning of the app. The discrepancies between the versions of the two operating systems led to changes in the onboarding process, as well as in the support of participants in working with the app. Additionally, these issues hindered the usability of the app and caused an unquantifiable yet noticeable loss-to-follow-up, thus presenting a key issue in the app assessment in the interpretation of RisCoin results [38,39]. Indeed, change management and, more specifically, app development across heterogeneous platforms has been shown to be among the biggest challenges for native app developers [40].

The latter challenge addresses a larger topic that needs to be considered when implementing mobile apps in the context of research. To be noted, the additional personnel resources are needed by the study team to support the participants in working with the app. While the personnel for the app onboarding during the enrollment was planned prior to recruitment, the continuous participant support via the app, via e-mail, and on-site had to be adjusted to the high number of support demands, especially following an update. Thus, we encourage future app-based studies to consider the personnel and resource costs associated with participants’ support for the full duration of the study.

### 4.3. Benefits of an App-Based Study Design

#### 4.3.1. Agility and Flexibility

As the recommendations for testing and vaccinations were being adapted during the pandemic, the app allowed us to adjust the short questionnaire, respectively. Since the questionnaire was consistently available to participants, the benefit of agility extends to participants as well. This facilitated the completeness and timeliness of data and may have reduced the recall bias [10]. Studies on app-based reporting by patients further highlight the effect of patient empowerment [41,42,43,44].

Several activities at the LMU University Hospital have also provided evidence for the flexibility of app-based data collection. The concurrently developed open-source prototype system for data acquisition in clinical studies has been thus far implemented in four clinical studies (including >1600 participants from different patient cohorts) [44].

#### 4.3.2. Facilitating Data Collection and Overview

Unlike paper-based data collection, the app enabled us to collect, request, or send queries to participants at any given time. This feature proved particularly useful during the data verification or when sending out reminders [45]. Although we encountered some technical issues, the self-reported data demonstrated high consistency, which allows us to explore longitudinal developments in cohort data in future RisCoin analyses.

Previous studies examining paper- vs. online-based data collection demonstrated a significantly higher adherence and preference of participants to the online-based option [3,46,47]. Mobile- or app-based methods generally facilitate prompt data collection and higher rates of data validity [5,48,49,50]. However, the evidence on study participants’ preferences and adherence to different modes of communication and self-reporting within a single study remains inconclusive [10]. As we did not explore comparative preferences for paper- vs. app-based data collection within RisCoin, we are only able to report on the utilization and usability of the chosen app-based mode of bidirectional communication and data self-reporting.

### 4.4. Strengths and Limitations of Our Research

Our study’s strengths include assessing app implementation and adherence in a cohort of nearly 4000 participants over more than a year of follow-up. We analyzed participant characteristics using data from various sources and concerning demographics, occupation, health, and lifestyle. This delivered an insightful view of potential target groups for app-based clinical studies. We also detailed the app’s technical development and implementation, offering a blueprint for future research. Furthermore, we discussed the experiences, advantages, and drawbacks from the study team’s perspective. These could contribute to understanding the role of digital tools in research.

Several limitations need to be taken into account when interpreting our results. Firstly, the study protocol did not foresee collecting quantifiable data on participants’ satisfaction with the app. Instead, our focus prioritized participants’ adherence to the app [40]. Secondly, the small number of patients with IBD could have limited the strength of associations within this sub-cohort. Third, the choice of threshold values with 12 submissions of a short questionnaire (75th percentile of the cohort) may be critical. However, there are no standards available for measuring adherence of a participant in using a digital tool. Lastly, some participants, particularly from the HCWs sub-cohort, may have dropped out of the study as a consequence of a terminated contract with the LMU University Hospital, parental leave, or retirement. As the study team did not cooperate with the Human Resources department (due to data privacy considerations), we were not being informed of relevant personnel fluctuations that may have resulted in loss to follow-up.

Considering limitations of the technical implementation, the app did not include the option for sending push notifications due to concerns for the privacy of the data. As the setup of push notifications requires the app to provide access to third parties, the development and the study team decided against exposing the app and the participants’ data to any risks arising from third-party access. This technical limitation may have led to some participants missing relevant information and, subsequently, potentially impeding their adherence to the app and the study protocol. Additionally, we note that this limitation has certainly led to the study team not being able to correct implausible or conflicting data reported by participants, i.e., contradictory vaccination information, as many queries on that matter remained unresolved and, consequently, led to missing data points in the dataset.

## 5. Conclusions

Implementing app-based cohort management is a feasible solution, offering flexibility and securing two-way anonymous communication. Adherence to app-based communication and data collection was remarkably high among patients, nurses, and administrative staff and increased significantly with participants’ age, suggesting that older participants are a reliable sub-cohort in app-based studies. Despite technical limitations requiring continuous support, the RisCoin study app proved effective for collecting longitudinal self-reported health data during the COVID-19 pandemic, even after irreversible anonymization. Our findings underscore the utility of mobile apps for epidemiological monitoring, considering adequate technical infrastructure and user support. Considering the cohort characteristics before and during app development, particularly age distribution and occupational background, may help identify barriers and facilitators to study protocol and app adherence, as well as indicate appropriate strategies to increase participants’ motivation for their active longitudinal involvement in the study.

## Figures and Tables

**Figure 1 idr-17-00088-f001:**
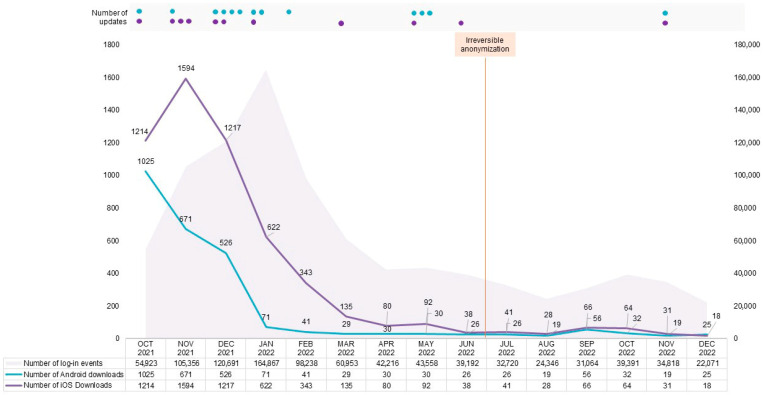
Number of app downloads and login events alongside number and timing of Android and iOS updates in the period October 2021–December 2022. Number of app downloads from the Play Store and Apple Store alongside cumulative frequency of login events were presented. Dots in respective colors indicate number of Android and iOS updates during October 2021–December 2022, as follows: October 2021: 1 (Android), 1 (iOS); November 2021: 1 (Android), 3 (iOS); December 2021: 4 (Android), 2 (iOS); January 2022: 2 (Android), 1 (iOS); February 2022: 1 (Android), 0 (iOS); March 2022: 0 (Android), 1 (iOS); April 2022: no updates; May 2022: 3 (Android), 1 (iOS); June 2022: 0 (Android), 1 (iOS); July–October 2022: no updates; November 2022: 1 (Android), 1 (iOS).

**Figure 2 idr-17-00088-f002:**
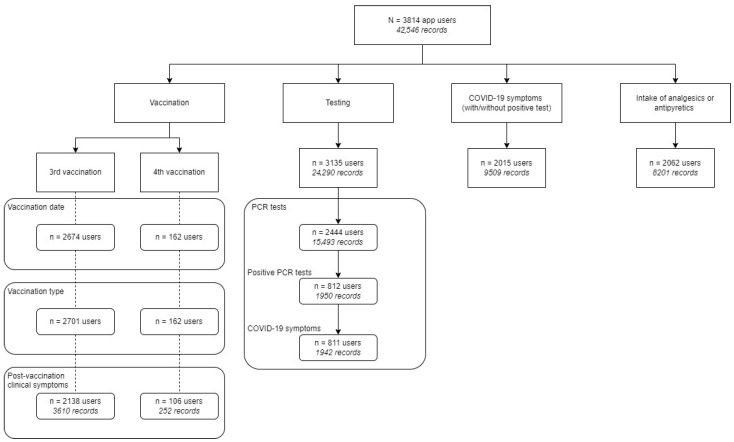
User statistics presenting frequency and type of submissions by RisCoin participants via the short questionnaire and hotline module in the app. Data entries were obtained from CentraXX app user statistics, including frequencies and purposes of login events via a short questionnaire and bidirectional communication in the hotline module. The short questionnaire was updated with questions about booster vaccines (3rd and 4th vaccination) according to pandemic developments. The cumulative frequency of key components in participant monitoring, such as vaccination status, PCR-test results, SARS-CoV-2 infection symptoms, and medication usage, was presented.

**Figure 3 idr-17-00088-f003:**
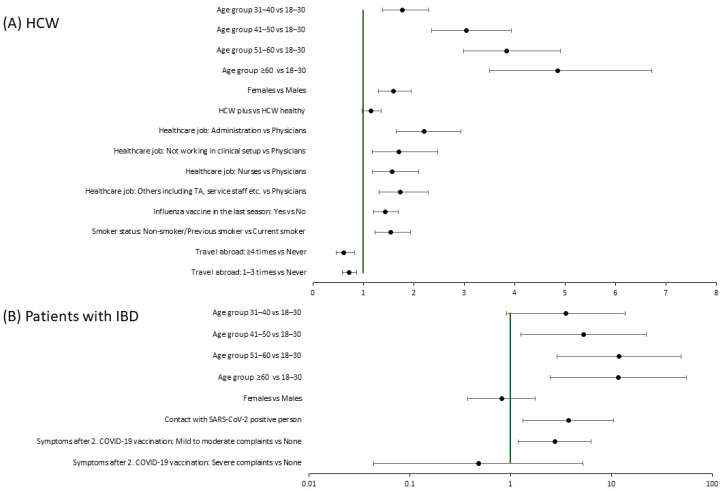
Factors associated with adherent usage of RisCoin study app among HCWs and patients with IBD. (**A**) Factors associated with adherent usage of RisCoin study app among HCWs. (**B**) Factors associated with adherent usage of RisCoin study app among patients with IBD. Odds ratios (OR) with 95% confidence intervals (95% CI) were obtained from the final multivariable logistic regression adjusted for gender, age, health status of health care workers with or without underlying disease (HCW-plus, HCW-healthy, respectively) to determine factors associated with adherent usage of RisCoin study app (≥12 completed questionnaires during the study period). *p*-values were determined using the Wald Chi-square test to assess the significance of the odds ratio (OR). (1) HCW-healthy includes all health care workers, who did not report any underlying disease or any medication intake at enrollment. (2) HCW-plus includes all health care workers, who reported regular medication intake or at least one of the following underlying diseases: cardiovascular disease, chronic pulmonary disease, diabetes mellitus, thyroid dysfunction, hypothyroidism, chronic renal disease, renal insufficiency, chronic hepatic or gastrointestinal disease, chronic neurological disease or disorder, cancer, transplantation, chronic hematological disease, rheumatological disease, or primary immunodeficiency disorders. (3) PSQ score: “Perceived Stress Questionnaire” Score; Participants filled out the 20-item validated version of the PSQ within the baseline questionnaire for RisCoin; we use the score of 33 as benchmark per the validation study of Fliege et al., where the score of 33 was determined for the healthy German population [6,14]. Abbreviations: BMI, body mass index; GI, gastrointestinal; HCWs: health care workers; IBD: inflammatory bowel disease; OR, odd ratio; PCR, polymerase chain reaction; TA: technical assistants.

**Table 1 idr-17-00088-t001:** Usage of RisCoin study app indicated by frequency of short questionnaire entries among HCWs (N = 3479) and patients with IBD (N = 143).

	RisCoin Sub-Cohorts							
	HCW				Patients with IBD			
	Non-Adherent Users N = 2521 (73%)	Adherent Users N = 958 (27%)	Total N = 3479	*p*-Value	Non-Adherent Users N = 85 (60%)	Adherent Users N = 58 (40%)	Total N = 143	*p*-Value
**RisCoin HCWs health status**				**<0.001 ^1^**	N/A	N/A		
HCW-plus ^2^	1125 (68%)	539 (32%)	1664 (48%)		N/A	N/A		
HCW-healthy ^3^	1396 (77%)	419 (23%)	1815 (52%)		N/A	N/A		
**Gender**				**<0.001 ^1^**				0.960 ^1^
Males	706 (80%)	173 (20%)	879 (25%)		48 (59%)	33 (41%)	81 (57%)	
Females	1808 (70%)	784 (30%)	2592 (75%)		37 (60%)	25 (40%)	62 (43%)	
**Age groups**				**<0.001 ^1^**				**0.029 ^1^**
18–30	873 (86%)	137 (14%)	1010 (29%)		22 (85%)	4 (15%)	26 (18%)	
31–40	678 (78%)	191 (22%)	869 (25%)		22 (61%)	14 (39%)	36 (25%)	
41–50	424 (66%)	218 (34%)	642 (18%)		16 (59%)	11 (41%)	27 (19%)	
51–60	417 (59%)	292 (41%)	709 (20%)		16 (46%)	19 (54%)	35 (24%)	
>60	129 (52%)	120 (48%)	249 (7%)		9 (47%)	10 (53%)	19 (13%)	
**BMI in 4 categories** ^4^				**0.004 ^1^**				0.188 ^1^
Underweight	72 (71%)	29 (29%)	101 (3%)		3 (75%)	1 (25%)	4 (3%)	
Normal weight	1602 (74%)	565 (26%)	2167 (62%)		45 (63%)	26 (37%)	71 (50%)	
Pre-obesity	610 (72%)	235 (28%)	845 (24%)		24 (48%)	26 (52%)	50 (35%)	
Obesity all classes	236 (65%)	129 (35%)	365 (10%)		13 (72%)	5 (28%)	18 (13%)	
**Part-time employment**				**<0.001 ^1^**				0.876 ^1^
Yes	737 (67%)	363 (33%)	1100 (32%)		20 (61%)	13 (39%)	33 (23%)	
No	1782 (75%)	594 (25%)	2376 (68%)		65 (59%)	45 (41%)	110 (77%)	
**Education**				<0.001 ^1^				0.326 ^1^
Middle school diploma	217 (63%)	128 (37%)	345 (10%)		19 (66%)	10 (34%)	29 (21%)	
High school diploma	628 (74%)	220 (26%)	848 (25%)		15 (71%)	6 (29%)	21 (15%)	
Completed apprenticeship	481 (67%)	239 (33%)	720 (21%)		18 (56%)	14 (44%)	32 (23%)	
University degree	1170 (77%)	353 (23%)	1523 (44%)		29 (51%)	28 (49%)	57 (41%)	
**Healthcare occupation**				**<0.001 ^1^**				0.207 ^1^
Nurses	592 (71%)	246 (29%)	838 (24%)		4 (80%)	1 (20%)	5 (3%)	
Physicians	482 (83%)	100 (17%)	582 (17%)		2 (67%)	1 (33%)	3 (2%)	
Administration	452 (65%)	245 (35%)	697 (20%)		0 (0%)	2 (100%)	2 (1%)	
Others including TA, service staff	783 (73%)	288 (27%)	1071 (31%)		1 (25%)	3 (75%)	4 (3%)	
Clinical scientists not working in patient care	199 (73%)	74 (27%)	273 (8%)		78 (60%)	51 (40%)	129 (90%)	
**Avoiding special foods i.e., due to allergy or intolerance**				**0.003 ^1^**				0.804 ^1^
Yes	148 (64%)	83 (36%)	231 (7%)		19 (58%)	14 (42%)	33 (23%)	
No	2373 (73%)	875 (27%)	3248 (93%)		66 (60%)	44 (40%)	110 (77%)	
**Participants with direct patient contact**				**<0.001 ^1^**				0.487 ^1^
Yes	1631 (75%)	535 (25%)	2166 (62%)		6 (50%)	6 (50%)	12 (8%)	
No	887 (68%)	422 (32%)	1309 (38%)		79 (60%)	52 (40%)	131 (92%)	
**COVID-19-related questions**								
**Contact with confirmed SARS-CoV-2 infected person**				**0.001 ^1^**				**0.049 ^1^**
Yes	1232 (75%)	410 (25%)	1642 (47%)		11 (42%)	15 (58%)	26 (18%)	
No or Unknown	1276 (70%)	544 (30%)	1820 (53%)		74 (63%)	43 (37%)	117 (82%)	
**Intensity of clinical symptoms after the 2nd COVID-19 vaccination**				0.094 ^1^				0.074 ^1^
None	1052 (73%)	395 (27%)	1447 (43%)		56 (65%)	30 (35%)	86 (61%)	
Mild or moderate complaints	1144 (71%)	462 (29%)	1606 (47%)		23 (46%)	27 (54%)	50 (36%)	
Severe complaints	258 (77%)	77 (23%)	335 (10%)		3 (75%)	1 (25%)	4 (3%)	
**Vaccinated against influenza during the last flu season**				**<0.001 ^1^**				**0.040 ^1^**
Yes	1231 (68%)	571 (32%)	1802 (53%)		43 (51%)	41 (49%)	84 (63%)	
No	1224 (77%)	375 (23%)	1599 (47%)		34 (69%)	15 (31%)	49 (37%)	
**Pollen allergy**				**<0.001 ^1^**				0.908 ^1^
Yes	735 (69%)	337 (31%)	1072 (31%)		25 (58%)	18 (42%)	43 (30%)	
No	1776 (74%)	621 (26%)	2397 (69%)		58 (59%)	40 (41%)	98 (70%)	
**Smoking status (consumption of tobacco products e-cigarettes hookah pipe)**				**<0.001 ^1^**				0.209 ^1^
Current smoker	475 (79%)	128 (21%)	603 (18%)		18 (69%)	8 (31%)	26 (19%)	
Non-smoker/Previous smoker	2004 (71%)	822 (29%)	2826 (82%)		63 (56%)	50 (44%)	113 (81%)	
**Self-perceived stress, PSQ > 33** ^5^				0.108 ^1^				0.992 ^1^
Yes	1401 (74%)	505 (26%)	1906 (55%)		43 (59%)	30 (41%)	73 (52%)	
No	1107 (71%)	451 (29%)	1558 (45%)		40 (59%)	28 (41%)	68 (48%)	

Results were presented in frequency (*n*) and row percentage (%) for the utilization frequency of RisCoin study app indicated by the number of entries through the short questionnaire: Adherent usage with ≥12 entries (75th percentile of the total cohort). Missing data, resulting from self-reporting, was evident as differences between the sums of subcategories and the total *n*. (^1^) *p*-values obtained by Pearson’s Chi-square test to indicate significant differences in the proportion of non-adherent usage (or adherent usage) across different categories of a specific factor. *p*-values ≤ 0.05 were considered statistically significant. (^2^) HCW-plus includes all health care workers, who reported regular medication intake or at least one of the following underlying diseases: cardiovascular disease, chronic pulmonary disease, diabetes mellitus, thyroid dysfunction, hypothyroidism, chronic renal disease, renal insufficiency, chronic hepatic or gastrointestinal disease, chronic neurological disease or disorder, cancer, transplantation, chronic hematological disease, rheumatological disease, or primary immunodeficiency disorders. (^3^) HCW-healthy includes all health care workers, who did not report any underlying disease or any medication intake at enrollment. (^4^) BMI categories were obtained by applying the WHO criteria [16]. (^5^) PSQ score: “Perceived Stress Questionnaire” Score; Participants filled out the 20-item validated version of the PSQ within the baseline questionnaire for RisCoin; we use the score of 33 as benchmark per the validation study of Fliege et al., where the score of 33 was determined for the healthy German population [6,14]. Abbreviations: HCWs: health care workers, IBD: inflammatory bowel disease, PSQ: perceived stress questionnaire.

## Data Availability

The data can be made available upon reasonable request.

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
