# Peer review of "Implementation and Adherence of a Custom Mobile Application for Anonymous Bidirectional Communication Among Nearly 4000 Participants: Insights from the Longitudinal RisCoin Study ^[Author-notes fn2-idr-17-00088]^"

_2036-7449, 2025, doi:10.3390/idr17040088_

Round 1
Reviewer 1 Report
Comments and Suggestions for Authors
This is an interesting piece of research, but the discussion section could be further developed, particularly by highlighting the relevance of the study and proposing future research avenues.
We encourage you to explore and integrate additional literature to strengthen and enrich your contribution
Bain, E. E., Shafner, L., Walling, D. P., Othman, A. A., Chuang-Stein, C., Hinkle, J., & Hanina, A. (2017). Use of a novel artificial intelligence platform on mobile devices to assess dosing compliance in a phase 2 clinical trial in subjects with schizophrenia. JMIR mHealth and uHealth, 5(2), e7030.
Menvielle, L., Ertz, M., François, J., & Audrain-Pontevia, A. F. (2022). User-Involved Design of Digital Health Products. In Revolutions in Product Design for Healthcare: Advances in Product Design and Design Methods for Healthcare (pp. 1-19). Singapore: Springer Singapore.
Duan, H., Wang, Z., Ji, Y., Ma, L., Liu, F., Chi, M., ... & An, J. (2020). Using goal-directed design to create a mobile health app to improve patient compliance with hypertension self-management: development and deployment. JMIR mHealth and uHealth, 8(2), e14466.
Author Response
We thank all reviewers for the time and effort invested in reading and commenting our manuscript. The revision provided a much needed opportunity for us to enhance the quality of our paper by offering a broader background of the app implementation and study execution, clarifying the study aims and contextualizing the results into a broader perspective.
We have utilized the revision to implement changes throughout the whole length of the paper and our point-by-point responses highlight the corrections more specifically requested or suggested by the reviewers.
COMMENT:
This is an interesting piece of research, but the discussion section could be further developed, particularly by highlighting the relevance of the study and proposing future research avenues.
We encourage you to explore and integrate additional literature to strengthen and enrich your contribution
Bain, E. E., Shafner, L., Walling, D. P., Othman, A. A., Chuang-Stein, C., Hinkle, J., & Hanina, A. (2017). Use of a novel artificial intelligence platform on mobile devices to assess dosing compliance in a phase 2 clinical trial in subjects with schizophrenia. JMIR mHealth and uHealth, 5(2), e7030.
Menvielle, L., Ertz, M., François, J., & Audrain-Pontevia, A. F. (2022). User-Involved Design of Digital Health Products. In Revolutions in Product Design for Healthcare: Advances in Product Design and Design Methods for Healthcare (pp. 1-19). Singapore: Springer Singapore.
Duan, H., Wang, Z., Ji, Y., Ma, L., Liu, F., Chi, M., ... & An, J. (2020). Using goal-directed design to create a mobile health app to improve patient compliance with hypertension self-management: development and deployment. JMIR mHealth and uHealth, 8(2), e14466.
RESPONSE:
- Thank you for the encouraging review; Taking into account the suggestions by all reviewers, we have extended the discussion and provided more detail and contextualization on the relevant aspects of the results
Reviewer 2 Report
Comments and Suggestions for Authors
This manuscript constitutes a sub-investigation within the broader RisCoin research initiative, which examined risk determinants associated with COVID-19 vaccine inefficacy among healthcare personnel and individuals diagnosed with inflammatory bowel disease at LMU University Hospital in Munich, while this particular paper concentrates specifically on the creation, deployment, and utilization patterns observed with a customized mobile application designed to facilitate secure, anonymous, and bidirectional communication channels within the study population. The application served as a mechanism for longitudinal data acquisition, laboratory result distribution, and sustained participant involvement throughout the comprehensive 15-month observation period, demonstrating the integration of digital health technologies within traditional epidemiological research frameworks.
The research team documents impressive adoption rates whereby nearly 4,000 enrolled study subjects resulted in 91% submitting at least one questionnaire through the digital platform, and 28% that achieved the predetermined threshold for classification as "frequent" users, while predictive factors for intensive utilization, encompassing demographic variables such as age and gender, behavioral indicators including influenza vaccination history and smoking patterns, and occupational categories, underwent comprehensive analysis through logistic regression modeling techniques.
Several methodological and interpretative dimensions warrant improvmenet or better details before this manuscript can be published.
- The definition of "frequent app user" as someone who submitted 12 or more short questionnaires seems arbitrary, even though it represents the 75th percentile. The authors need to explain more: how were these 12 entries spread out over time? And did this pattern match with scheduled study visits or blood draws? Without this context, the definition lacks scientific basis. It would help to show the full range of how many questionnaires each person completed and see if there were different patterns like consistent users, early dropouts, or occasional participants. Advanced statistical methods might work better than just dividing users into frequent versus non-frequent groups.
- Another problem is not paying enough attention to people who didn't use the app or dropped out: about 9% of enrolled participants never submitted any questionnaires. The paper mentions how this group included more full-time doctors and male healthcare workers but doesn't explain much: were these people also less likely to complete other parts of the study like blood tests? Did technical problems affect certain groups more? Understanding these barriers would make the paper more valuable for digital health research, where knowing why people don't participate matters as much as studying those who do.
- The multivariable regression analysis is well presented, but findings are not sufficiently discussed. For example, why are nurses and administrative staff more engaged than physicians? Does this reflect differing availability, motivation, or institutional role in health information systems? Similarly, gender effects are noted but not interpreted, despite ample literature showing higher digital health engagement among women in both clinical and community settings (Becker 2017).
Finally, the claim of “irreversible anonymisation” is not that clear, in the context of the methodology of the paper. A clearer timeline of pseudonymisation, de-identification, and any residual re-identification risk would be essential to let readers understand and reassure ethics reviewers.
Finally, the Discussion might benefit from a broader digital-health perspective that succeed in framing the observed adherence curves alongside large remote-study retention data (Pratap et al., 2020) and known usability barriers in mobile epidemiology (Fischer & Kleen, 2021); this would help contextualise the 27–40 % frequent-user rate. The authors could also consider the emerging role of conversational agents as a complement to app-based data capture and feedback, particularly for vaccine communication and health-literacy support (Cosma et al., 2025).
https://pubmed.ncbi.nlm.nih.gov/39615346/
https://pubmed.ncbi.nlm.nih.gov/33480850/
https://pubmed.ncbi.nlm.nih.gov/32128451/
Author Response
We thank all reviewers for the time and effort invested in reading and commenting our manuscript. The revision provided a much needed opportunity for us to enhance the quality of our paper by offering a broader background of the app implementation and study execution, clarifying the study aims and contextualizing the results into a broader perspective.
We have utilized the revision to implement changes throughout the whole length of the paper and our point-by-point responses highlight the corrections more specifically requested or suggested by the reviewers.
COMMENT:
The definition of "frequent app user" as someone who submitted 12 or more short questionnaires seems arbitrary, even though it represents the 75th percentile. The authors need to explain more: how were these 12 entries spread out over time? And did this pattern match with scheduled study visits or blood draws? Without this context, the definition lacks scientific basis. It would help to show the full range of how many questionnaires each person completed and see if there were different patterns like consistent users, early dropouts, or occasional participants. Advanced statistical methods might work better than just dividing users into frequent versus non-frequent groups.
RESPONSE:
- Thank you for suggesting a more detailed description of our approach.
A cut-off of 12 entries in the app was selected because it represents the 75th percentile of the number of entries per individual within the Riscoin cohort. First, in the absence of standardized thresholds in the literature, selecting a cut-off based on the distribution within the cohort is a commonly accepted methodological approach. Second, the number of entries over time, including follow-up duration, is shown in Figure 1. This figure demonstrates that the majority of participants used the app continuously for 3 months after enrollment. On average, participants submitted approximately one entry per week, indicating adherence to app use during the first 12 weeks after enrollment. Based on these observations, we considered 12 entries a reasonable threshold to distinguish between adherent and non- adherent users. Nevertheless, this approach has been acknowledged as a limitation of the study and discussed accordingly in the manuscript.
In our opinion, applying other statistical methods to evaluate factors associated with an increasing number of entries would not be an appropriate approach for this cohort, because the critical observation period encompasses the first 3 months after enrollment, particularly after the third vaccination. The cut-off of 12 entries corresponds to this period, during which infections and symptoms after third vaccination were predominantly observed. The observation for this consideration was additionally presented in the supplementary file (Table 1 and the now added Table 2).
Thank you very much for paying attention to the categorization of participaint according to the frequency of using app. You are right, if so, it should be conisdered consistent users, early dropouts, or occasional participants. After the 6-month mark, participants were instructed to complete the questionnaire only in the event of infection or vaccination. Consequently, the classification of "occasional participants" applies to all cases beyond the 6-month period. From this reason, we changed the category of users into: adherent and non-adherent users.
We are aware of our approach and its limitation, so that we have demontrated it as one of our limitation in the discussion.
- An additional aspect needs to be taken into account, namely the fluctuation rate of the HCW leaving the hospital and, with that, also the study. The study team has not been specifically made aware of the HCW with terminated contracts, which respectively prevented the team from any follow-up attempts.
COMMENT:
- Another problem is not paying enough attention to people who didn't use the app or dropped out: about 9% of enrolled participants never submitted any questionnaires. The paper mentions how this group included more full-time doctors and male healthcare workers but doesn't explain much: were these people also less likely to complete other parts of the study like blood tests? Did technical problems affect certain groups more? Understanding these barriers would make the paper more valuable for digital health research, where knowing why people don't participate matters as much as studying those who do.
RESPONSE:
- Thank you for noting the need to elaborate further on the non-active user group. We have now added a more detailed description (lines #284-300) and included additional results on this group (Supplement Table 2). To further elaborate, the non-users group had donated the first blood sample, but did not come back for any further follow-up. Due to the study design and the complete anonymization, we only could get into individual contact with the participants via the app. There we can only speculate on the different reasons for the 9% non-users: staff fluctuation in the university hospital, problems with loading or working with the app, loss of intestest, lack of time (please refer to our reply to the previous comment as well).
COMMENT:
The multivariable regression analysis is well presented, but findings are not sufficiently discussed. For example, why are nurses and administrative staff more engaged than physicians? Does this reflect differing availability, motivation, or institutional role in health information systems? Similarly, gender effects are noted but not interpreted, despite ample literature showing higher digital health engagement among women in both clinical and community settings (Becker 2017).
RESPONSE:
- We have extended the discussion section on the gender-specific interpretation of the results (lines #385-392) as well as on the considerations regading the higher adherence of IBD patients to the app compared to HCW (lines #426-437). We have also provided an interpretation for the lower adherence of physicians compared to other occupational groups, (lines #439-468). We may hypothesize that the comparatively longer working day and week of physicians, particularly during the pandemic, may have resulted in this subcohort indeed having less time and capacity to follow the study protocol as intended, however we note that no specific questions were asked in this regard since the general adherence and functioning of the app were at the forefront of the primary objective. Still, physicians were expected to present higher adherence, especially since the context of a university hospital should highlight the understanding and appreciation for high-quality longitudinal data collection.
COMMENT:
Finally, the claim of “irreversible anonymisation” is not that clear, in the context of the methodology of the paper. A clearer timeline of pseudonymisation, de-identification, and any residual re-identification risk would be essential to let readers understand and reassure ethics reviewers.
RESPONSE:
- The concept of data pseudonymisation was presented in detail in basic paper on study design, to which we refer to in ref. 6 Koletzko, S.; Le Thi, T.G.; Zhelyazkova, A.; Osterman, A.; Wichert, S.P.; Breiteneicher, S.; Koletzko, L.; Schwerd, T.; Völk, S.; Jebrini, T.; et al. A prospective longitudinal cohort study on risk factors for COVID-19 vaccination failure (RisCoin): methods, procedures and characterization of the cohort. Clin. Exp. Med. 2023, doi:10.1007/s10238-023-01170-6. We have now included two additional sentences (lines #163-165)
- Additionally and to facilitate a better understanding of the timeline, we added a visualization of the timing of final anonymization to Figure 1
COMMENT:
Finally, the Discussion might benefit from a broader digital-health perspective that succeed in framing the observed adherence curves alongside large remote-study retention data (Pratap et al., 2020) and known usability barriers in mobile epidemiology (Fischer & Kleen, 2021); this would help contextualise the 27–40 % frequent-user rate. The authors could also consider the emerging role of conversational agents as a complement to app-based data capture and feedback, particularly for vaccine communication and health-literacy support (Cosma et al., 2025).
https://pubmed.ncbi.nlm.nih.gov/39615346/
https://pubmed.ncbi.nlm.nih.gov/33480850/
https://pubmed.ncbi.nlm.nih.gov/32128451/
RESPONSE:
- Additionally, we extended the section regarding travel behaviour as the positioning of these results into a broader perspective of uncertainty may help contextualize the general health awareness and respective behaviour of RisCoin participants during the COVID-19 pandemic and the therewith associated public health restricting measures to travel.
- We also thank the reviewer for the very helpful reference suggestions – the study by Fischer and Kleen has already been cited in the introduction (ref. 5).
Reviewer 3 Report
Comments and Suggestions for Authors
1 Authors need to confirm that all acronyms are defined before being used for the first time. For example, LMU, app in abstract.
2 From the abstract I don't really understand exactly what the study objectives are and how they correlate with the results. Please rethink the abstract by bringing the objectives correlated with the results and conclusions.
3 The background part must be completed. It is not sufficiently documented and argued if there are any similar ideas regarding the use of applications related to the study objectives.
The first objective is not clearly defined. I recommend a clear redefinition as primary aim and secondary 81
aim
4 Method:
a) In the data extraction and analysis chapter it is specified what software was used to process the data extracted from the application or form. With or without a license, owned by whom?
Results
In the first paragraph, I recommend authors to introduce the aims of this study
5. Discussion
I observed that for patients with IBD a higher perception of risk was shown. I believe that this area should be developed in discussions in which patients with increased attention to their health status have a higher perception of risk and therefore the chance of using specific tools for disease prevention and control. How does this aspect correlate with vaccination adherence?
6 The conclusions should be aligned with the redefinition of the study objectives in the background.
7 Otherwise the manuscript is good, well structured, the bibliographical references are well inserted.
8 The manuscript is scientifically good and the experimental design is appropriate to answer the research hypotheses
9 The study is comprehensive and well written. I recommend its publication.
Author Response
We thank all reviewers for the time and effort invested in reading and commenting our manuscript. The revision provided a much needed opportunity for us to enhance the quality of our paper by offering a broader background of the app implementation and study execution, clarifying the study aims and contextualizing the results into a broader perspective.
We have utilized the revision to implement changes throughout the whole length of the paper and our point-by-point responses highlight the corrections more specifically requested or suggested by the reviewers.
COMMENT:
1 Authors need to confirm that all acronyms are defined before being used for the first time. For example, LMU, app in abstract.
RESPONSE:
- Thank you for noting this; we have written out the word „application“ in the abstract. Please note that the name of the University Hospital has to be spelled out as LMU University Hospital as per the dean’s directive. We have made the necessary corrections in the revised version.
COMMENT:
2 From the abstract I don't really understand exactly what the study objectives are and how they correlate with the results. Please rethink the abstract by bringing the objectives correlated with the results and conclusions.
RESPONSE:
- Thank you for noting this issue. We have re-written the abstract, so that the objectives are now more related to the results and conclusions.
COMMENT:
3 The background part must be completed. It is not sufficiently documented and argued if there are any similar ideas regarding the use of applications related to the study objectives.
The first objective is not clearly defined. I recommend a clear redefinition as primary aim and secondary aim
RESPONSE:
- Thank you for indicating the opportunity to rephrase background and additionally clarify the study objectives. We have adapted the introduction accordingly (lines #85-87).
COMMENT:
4 Method:
a) In the data extraction and analysis chapter it is specified what software was used to process the data extracted from the application or form. With or without a license, owned by whom?
RESPONSE:
- Thank you very much for the recommendations. The sentence clarified license was inserted into lines #178-1
COMMENT:
Results
In the first paragraph, I recommend authors to introduce the aims of this study
RESPONSE:
- Thank you for the recommendation – we adapted the introduction section as suggested.
COMMENT:
Discussion
I observed that for patients with IBD a higher perception of risk was shown. I believe that this area should be developed in discussions in which patients with increased attention to their health status have a higher perception of risk and therefore the chance of using specific tools for disease prevention and control. How does this aspect correlate with vaccination adherence?
RESPONSE:
- We have extended the discussion section on the considerations regading the higher adherence of IBD patients to the app compared to HCW and provided a broader interpretation of the results regarding this subcohort (lines #426-437).
COMMENT:
6 The conclusions should be aligned with the redefinition of the study objectives in the background.
RESPONSE:
- Thank you for noting this issue – we have adapted the conclusion accordingly (lines #616-628).
Round 2
Reviewer 2 Report
Comments and Suggestions for Authors
Thank you for sharing the revised manuscript, this version has clearly improved in terms of structure, clarity, and alignment with previous comments. One point that might still requires clarification is the rationale for the ≥12 questionnaire cutoff used to define adherent users; as it currently stands, the choice of this threshold does not seem to be explained, the only given reason is that this is the highest quartile.
Other than that, my congratulations to the authors.
Author Response
Reviewer comment:
Thank you for sharing the revised manuscript, this version has clearly improved in terms of structure, clarity, and alignment with previous comments. One point that might still requires clarification is the rationale for the ≥12 questionnaire cutoff used to define adherent users; as it currently stands, the choice of this threshold does not seem to be explained, the only given reason is that this is the highest quartile.
Other than that, my congratulations to the authors.
Response:
Thank you for suggesting a more detailed description of our approach. We have provided an exhaustive descriptive of the reasoning behind the determined threshold of ≥12 entries (lines #226-234).
We acknowledged the 12-entry threshold for defining adherent users, which based on statistical considerations (75th percentile) and the observation that the majority consistently made one entry per week over three months, as a limitation of our study and discussed it accordingly in the “Strengths and Limitations” section.